# A practical fluorosulfonylating platform via photocatalytic imidazolium-based SO$_2$F radical reagent

Weigang Zhang[1,2], Heyin Li[1,2], Xiaojuan Li[1], Zhenlei Zou[1], Mengjun Huang[1], Jiyang Liu[1], Xiaochen Wang[1], Shengyang Ni[1], Yi Pan[1] & Yi Wang [1✉]

Sulfonyl fluorides are key components in the fields of chemical biology, materials science and drug discovery. In this line, the highly active SO$_2$F radical has been employed for the construction of sulfonyl fluorides, but the utilization of gaseous ClSO$_2$F as radical precursor is limited due to the tedious and hazardous preparation. Meanwhile, the synthesis of sulfonyl fluorides from inert SO$_2$F$_2$ gas through a fluorosulfonyl radical (·SO$_2$F) process has met with inevitable difficulties due to the high homolytic bond dissociation energy of the S(VI)-F bond. Here we report a radical fluorosulfonylation strategy for the stereoselective synthesis of alkenyl sulfonyl fluorides and functional alkyl sulfonyl fluorides with an air-stable crystalline benzimidazolium fluorosulfonate cationic salt reagent. This bench-stable redox-active reagent offers a useful and operational protocol for the radical fluorosulfonylation of unsaturated hydrocarbons with good yield and high stereoselectivity, which can be further transformed into valuable functional SO$_2$F moieties.

[1] State Key Laboratory of Coordination Chemistry, Jiangsu Key Laboratory of Advanced Organic Materials, School of Chemistry and Chemical Engineering, Nanjing University, Nanjing 210023, China. [2]These authors contributed equally:Weigang Zhang, Heyin Li. ✉email: yiwang@nju.edu.cn

The sulfur(VI) fluoride exchange (SuFEx) chemistry that rely on the unique reactivity–stability balance of high valent organosulfur has emerged as a promising topic for the next-generation click reaction[1]. Sulfonyl fluorides as the most widely used connective hubs of SuFEx click reaction have attracted enormous attention and find widespread applications in the fields of chemical biology[2–6], drug discovery[7–11] and materials science[12–16]. Methods have been developed for rapid construction of sulfonyl fluoride moiety, including the chloride-fluoride exchange of sulfonyl chlorides[17–19], $SO_2$ insertion/fluorination[20–23], electrophilic fluorination of thiols and anodic oxidative fluorination[24–27]. Compared with the above mentioned S-F bond formation, direct fluorosulfonylation would provide a concise and redox economic approach for $C-SO_2F$ bond formation. The sulfonyl fluoride building-block[28,29] including alkynylsulfonyl fluoride (SASF)[30], ethenesulfonyl fluoride (ESF), 1-bromoethene-1-sulfonyl fluoride (BESF) were used to access functionalized sulfonyl fluorides. The highly active $SO_2F$ radical has been recognized as unstable and inaccessible precursor until the observation of this species from the decomposition of fluorosulfonyl azide and the recent progress of photoinduced radical fluorosulfonylation using gaseous $ClSO_2F$[31–36]. However, the application of $N_3SO_2F$ and $ClSO_2F$ were limited by tedious and hazardous preparation. Sulfuryl fluoride ($SO_2F_2$) as abundant inflammable industrial feedstock could serve as economic sulfonyl fluoride source[1]. Sulfuryl fluoride derived fluorosulfonylating reagents are mainly electrophilic "$FSO_2^+$" synthons and have been employed for direct functionalization of different nucleophiles including organometallic reagents, phenols, amines, etc.[37–40]. However, the construction of diversified sulfonyl fluoride compounds was limited by single electrophilic reaction pattern and hindered multifunctionalization of $SO_2F_2$ and derivatives. In contrast, adopting a radical synthesis strategy can overcome the limitations of electrophilic fluorosulfonylation and expand the scope of application of sulfonyl fluoride. However, the generation of fluorosulfonyl radical ($\cdot SO_2F$) from inert $SO_2F_2$ gas has met with inevitable difficulties due to the relatively small magnetic/quadrupole moments and the high homolytic bond dissociation energy of the S(VI)-F bond (BDE = $90.5 \pm 4.3$ kcal/mol)[1] (Fig. 1a). Thus, the development of single electron transfer (SET) process of $SO_2F_2$ for radical fluorosulfonylation represents great challenge and in high demand.

A practical procedure for the bench-stable redox-active $\cdot SO_2F$ agent from inexpensive fluorine source would provide appropriate solution to the long-standing issue of radical fluorosulfonylation. The imidazolium sulfonate cationic salt that developed in our lab has been successfully applied for the activation of triflic acid and arylsulfonates to access $\cdot SO_2CF_3$ and ArS$\cdot$ radicals[41–43]. We speculated that the cationic benzimidazole salt could harness the highly electrophilic $SO_2F$ to forge a bench-stable redox-active (Het)N-$SO_2F$ reagent (Fig. 1c)[44,45]. The positive charge of the resulting benzimidazolium fluorosulfonate can be delocalized on both nitrogens. By the homolytic cleavage of the weak N–S bond (BDE ≈ 70 kcal/mol)[46], this cationic complex undergoes SET process to generate fluorosulfonyl radical (Fig. 1d). In this work, we synthesize a series of highly reactive radical fluorosulfonylating reagents IMSF (2a-2e), practical and air-stable crystalline salts for a sequential radical stereoselective fluorosulfonylation, hydrofluorosulfonylation and migratory $SO_2F$-difunctionalization of unsaturated hydrocarbon to construct a variety of functionalized sulfonyl fluoride compounds.

## Results

**Reaction optimization**. Our study began with *N*-methyl-*N*-(1-phenylvinyl)acetamide (**1a**) as the model substrate (Table 1). After extensive screening of conditions, we found that when using 2 equivalents of benzimidazolium sulfonate reangent (IMSF, **2a**, $E_{1/2}^{red} = -1.07$ V *vs* SCE), 2 mol% of 4CzIPN, 2.5 equivalents of $KH_2PO_4$ in DME (1 mL) under the irradiation of 60 W blue LEDs, the alkenyl sulfonyl fluoride product **3a** could be obtained in 62% yield with >20:1 *E/Z* ratio. Different benzimidazolium sulfonate **2b-2e** were then examined (Table 1). When imidazolium sulfonate reagent **2b** were used, the yield of **3a** was obtained in 71% yield and isolated yield is 65% (entry 2). Other arylimidazole heterocycle with electron withdrawing groups derived IMSF salts **2c** and **2d** furnished alkenyl sulfonyl fluoride **3a** in 58% and 64% yield, respectively (entries 3 and 4). The cationic reagent **2e** resulted in a lower conversion under irradiation, which may due to the relatively high negative reduction potential (entry 5). When Ir(ppy)$_3$ instead of 4CzIPN as photocatalyst, the yield of product **3a** was slightly reduced (entry 6). When using other reaction solvents (Supplementary Table S1), the yield of desired product **3a** has significantly decreased and obtained in a low yield (entry 7). The yield of alkenyl sulfonyl fluoride was reduced in the absence of $KH_2PO_4$ because of the hindered $\alpha$-hydrogen elimination process (entry 8). In addition, control experiments suggested that photocatalyst, and light irradiation are all crucial to the reaction (entries 9–10).

**Substrate scope with respect to the radical alkenyl sulfonyl fluoride reaction**. With the optimized reaction conditions in hand, we next examined the generality of this transformation with different alkenes. Using 2 mol% of 4CzIPN, IMSF salt **2b** (2.0 equiv), and $KH_2PO_4$ (2.5 equiv) at ambient temperature, a range of alkenes underwent radical fluorosulfonylation with high efficiency. As shown in Fig. 2, 1,1-disubstituted alkenes with methyl, aryl, ester, amide groups afforded the desired products (**3a-3l**) in moderate to good yields with high regio- and stereoselectivity (*E:Z* > 20:1). Styrene with different substituents including halides, alkyl, ester afforded the desired products (**3n-3r, 3 v**) in moderate to excellent yields and high regio- and stereoselectivity (*E:Z* > 20:1). In addition, 1,2-dihydronaphthalene (**3 s**), 2-vinylthiophene (**3t**), 2-vinylpyridine (**3 u**) could all be smoothly fluorosulfonylated with $FSO_2$ radical. Moreover, natural products derivatized olefin involving cholesterol and estrone (**3w-3x**) can also be tolerated under the mild photocatalytic conditions and obtained the corresponding alkenyl sulfonyl fluoride in moderate yields. The selective preparation of *E*-alkenyl sulfonyl fluoride has been readily accessible. Then we try to control the reaction conditions to achieve the synthesis of thermodynamically less favorable *Z*-alkenyl sulfonyl fluoride. By variation of the reaction conditions (see Supplementary Table S5), we have extended this radical fluorosulfonylation protocol to achieve *Z*-alkenyl sulfonyl fluoride. Styrene with different substituents including halides, alkyl group afforded the desired products (**4b-4e, 4f-4g**) in moderate to good yields. Bioative DL-menthol and bexarotene derived alkenes could afford the desired alkenyl sulfonyl fluoride in moderate yields (**4h-4i**).

**Substrate scope with respect to the radical hydrofluorosulfonylation reaction**. The late stage functionalization of sulfonyl fluoride has been unearthed by Sharpless lab in 2014 with the development of SuFEx chemistry. Along this line, this radical fluorosulfonylation protocol was applied to the late-stage modification of complex molecules[47,48]. Using 2 mol% of the iridium catalyst, 1,4-cyclohexadiene as hydrogen donor (1.5 equiv), and IMSF salt **2** (2.0 equiv) at ambient temperature, unactivated terminal alkenes underwent a radical fluorosulfonylation process to product corresponding alkylsulfonyl fluoride with good regioselectivity (Fig. 3). Terminal alkenes bearing amide and ester functionalities obtained the desired alkylsulfonyl fluorides in

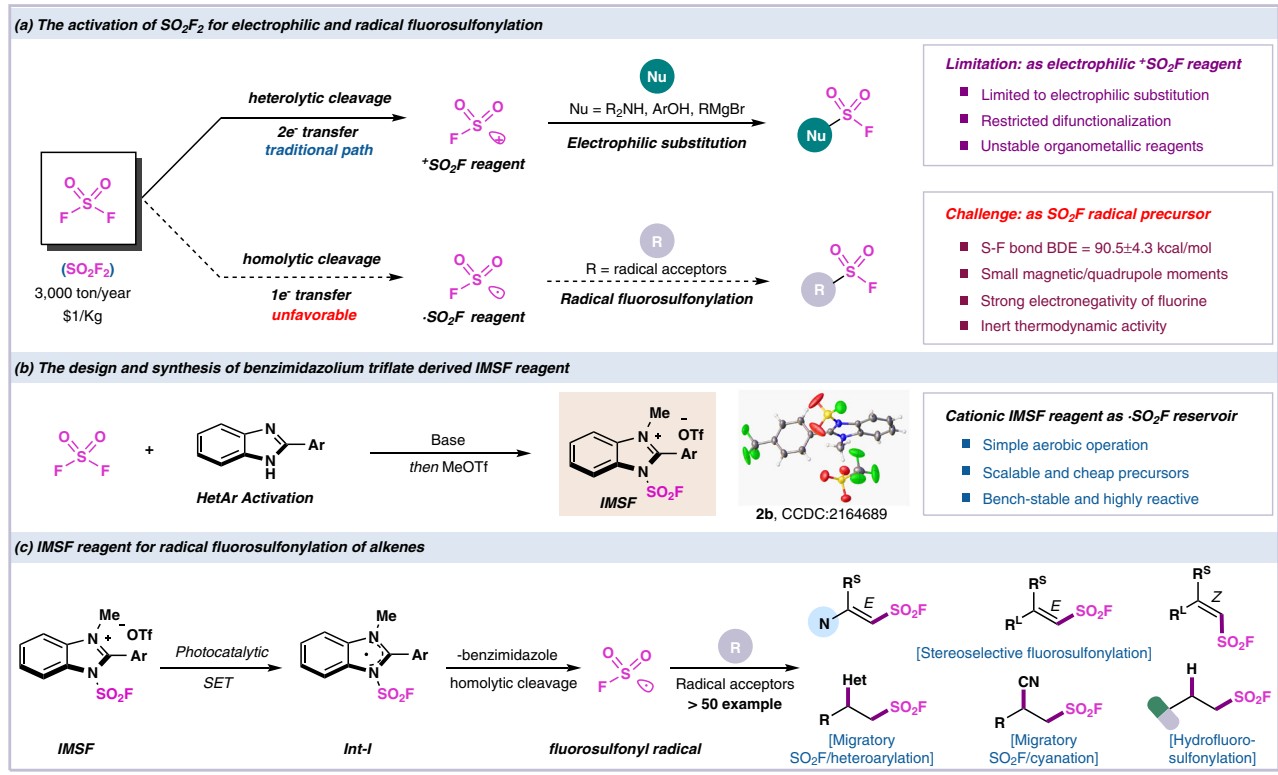

**Fig. 1 Origin of the reaction design. a** The activation of SO₂F₂ for electrophilic and radical fluorosulfonylation. **b** The design and synthesis of benzimidazolium triflate derived IMSF reagent. **c** This chemistry: Cationic IMSF reagent for radical fluorosulfonylation of alkenes.

## Table 1 Optimization of the reaction conditions.

| Entry | Variation from the conditions | Yield of 3aᵃ/% | E:Z of 3aᵇ |
|---|---|---|---|
| 1 | None | 62 | >20:1 |
| **2** | **2b instead of 2a** | **71 (65)ᶜ** | **>20:1** |
| 3 | 2c instead of 2a | 58 | >20:1 |
| 4 | 2d instead of 2a | 64 | >20:1 |
| 5 | 2e instead of 2a | 16 | >20:1 |
| 6 | Ir(ppy)₃ instead of 4CzIPN | 59 | >20:1 |
| 7 | Other solvents instead of DME | 0–41% | >20:1 |
| 8 | w/o KH₂PO₄ | 45 | >20:1 |
| 9 | w/o 4CzIPN | 0 | — |
| 10 | In the darkness | 0 | — |

ᵃ Yield determined by gas chromatography (GC) using dodecane as an internal standard; ᵇ The Z/E ratio was determined by ¹H NMR and GC; ᶜ Isolated yield.

moderate to good yields (**6a-6c, 6e, 6 f, 6h-6k**). Oxyalkyl-substituted alkenes also furnished the corresponding SO₂F adducts (**6 g**). In addition, IMSF reagent **2a** were employed in intramolecular cyclization process with diallyl sulphonamides to afford the corresponding sulfonyl fluoride product (**6d**).

**Substrate scope with respect to migration fluorosulfonylation reaction.** With slight variation of the optimized conditions[49,50], we expanded the scope of this radical fluorosulfonylation protocol to difunctionalization of unactivated olefins. Using a heteroaryl-substituted unsaturated tertiary alcohol, the distal migration induced by fluorosulfonyl radical proceeded smoothly in a chemoselective fashion. Fluorosulfonyl radical were susceptible to the

reaction conditions for achieving the corresponding ketones in good to excellent yields (Fig. 4). The aryl groups with different electronic and steric characters were tolerated (**8a-8b**). Thiophene and furan functionalities could be compatible under the mild condition (**8c-8d**). Linear or cyclic alkyl substituted unsaturated tertiary alcohols could also get the difunctionalized sulfonyl fluoride products (**8f-8h**). Noteworthy, the distal migration cyanation of unsaturated tertiary alcohol mediated by SO₂F radical also proceeded smoothly to afford desired product in good yield. The aryl group with electron donating and electron withdrawing groups and furan can afford the desired products (**8j-8l**) in good yields.

**Synthetic applications and mechanistic studies.** The utility of the products with sulfonyl fluoride group was demonstrated (Fig. 5a–e). In the presence of NaHCO₃ and DBU, alkenylsulfonyl fluoride **3 m** can easily react with pyrazolone **9** to afford the sulfone product **10** in 50% yield[51,52]. In addition, sulfonyl fluoride **3 m** can also efficiently react with 1,3-cyclohexanedione **11** to generate the sultone **12** (Fig. 5a)[53]. It is well known that sulfonyl fluoride species can readily undergo various SuFEx reactions to connect other molecules[54]. Several SuFEx reactions of selective modifying the hydroxyl site of drugs were implemented. We tentatively tried the ligation of styrenesulfonyl fluoride **3 m** with estrone **13**[55], which can afford the desired product **14** in 60% yield (Fig. 5b). Then the ligation of lumacator intermediate derivative (**6 f**) and vitamin E (**15**) reacted smoothly to furnish the desired product **16** in good yield (Fig. 5c).

To gain insight into this reaction, several mechanistic experiments have been carried out. The radical trapping experiment using 2 equiv. of TEMPO resulted in the inhibition of the radical addition. Instead, TEMPO adduct **17** was detected

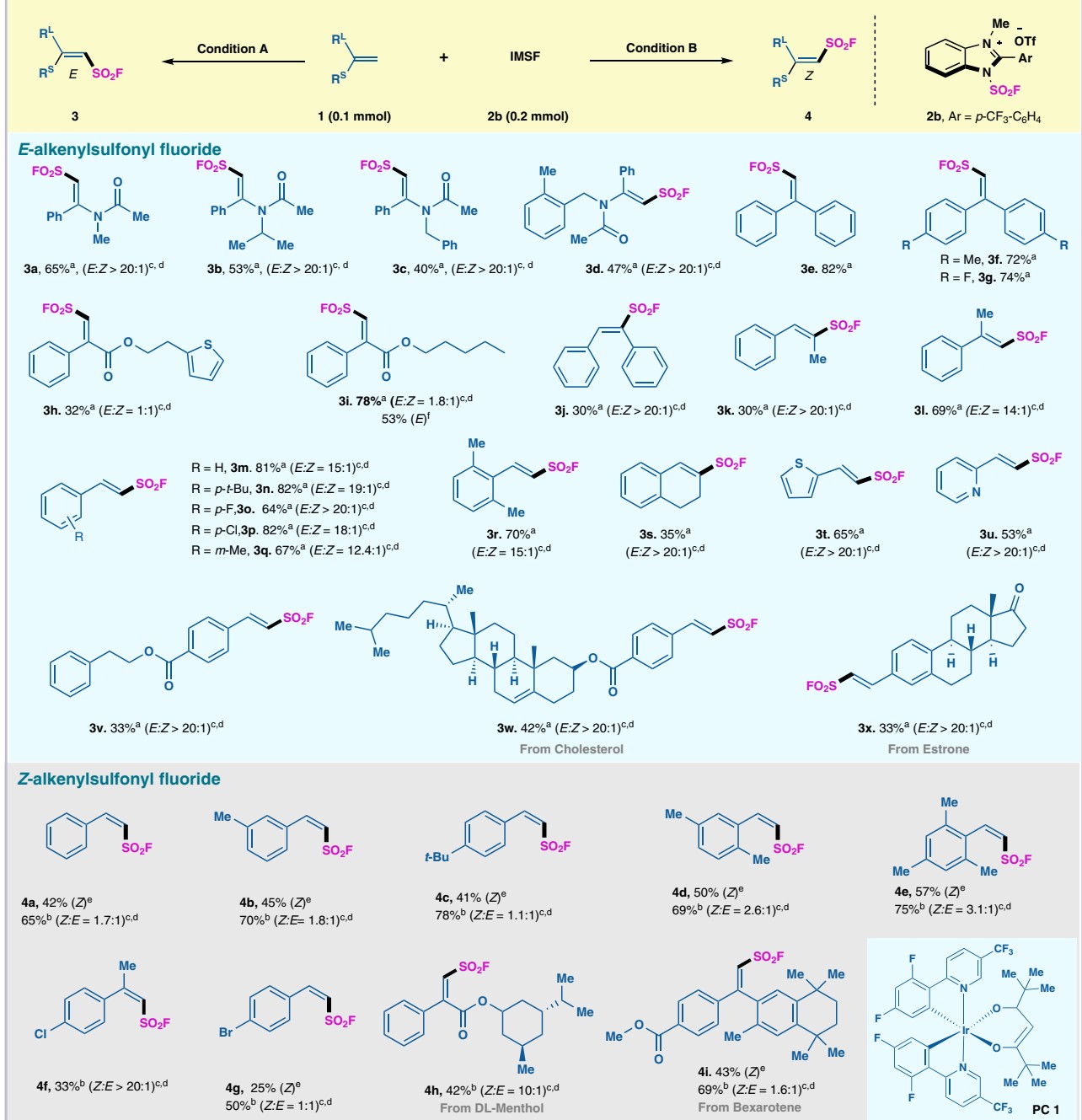

**Fig. 2 Substrate scope of the radical alkenylsulfonyl fluoride reaction.** [a.]Condition A: all reactions were carried out with olefins **1**, IMSF salt **2b** (0.20 mmol, 2 equiv), 4CzIPN (2 mol%), KH$_2$PO$_4$ (0.25 mmol, 2.5 equiv) in DME (1.0 mL) under Ar and 60 W blue LEDs. [b.]Condition B: all reactions were carried out with olefins **1**, IMSF salt **2b** (0.20 mmol, 2 equiv), PC 1 (2 mol%), KH$_2$PO$_4$ (0.25 mmol, 2.5 equiv) in EA:DME = 4:1 (1.0 mL) under Ar and 60 W blue LEDs, overnight, then 1.5 mL of acetonitrile containing Ir{[dF(CF$_3$)ppy]$_2$(dtbbpy)}PF$_6$ (2 mol%) was added to react about 12 h. [c.]The E/Z ratio was determined by $^{19}$F NMR. [d.]The E/Z ratio was determined by $^1$H NMR. [e.]Isolated yield.

in HRMS (Fig. 5d). The radical clock experiment was carried out using cyclopropyl styrene **1**. Under the standard conditions with MgCl$_2$ additive, the ring-opened product **19** can be obtained in 11% isolated yield (Fig. 5e). Thus, fluorosulfonyl radical (·SO$_2$F) intermediates are possibly involved in reaction. In addition, the treatment of E-alkenylsulfonyl fluoride **3n** with iridium catalyst in the absence of IMSF reagent furnished the Z-alkenylsulfonyl fluoride **4c** in 43% yield and recovered **3n** in 48% yield (Fig. 5f). This control experiment showed that the generation of Z-alkenylsulfonyl fluoride probably underwent an olefin isomerization process[56].

## Discussion

In summary, we have described an air-stable redox-active imidazolium fluorosulfonate reagent IMSF. A key design feature of this radical fluorosulfonylating reagent is the cationic nature, which favors the stepwise formation of fluorosulfonyl radical (·SO$_2$F) via a SET reduction process under photocatalytic conditions. This SO$_2$F radical reservoir could react with various alkenes to produce alkenyl sulfonyl fluoride, alkylsulfonyl fluoride, and migratory fluorosulfonylating products. Further studies of this highly reactive and air-stable solid reagent are underway in our laboratory.

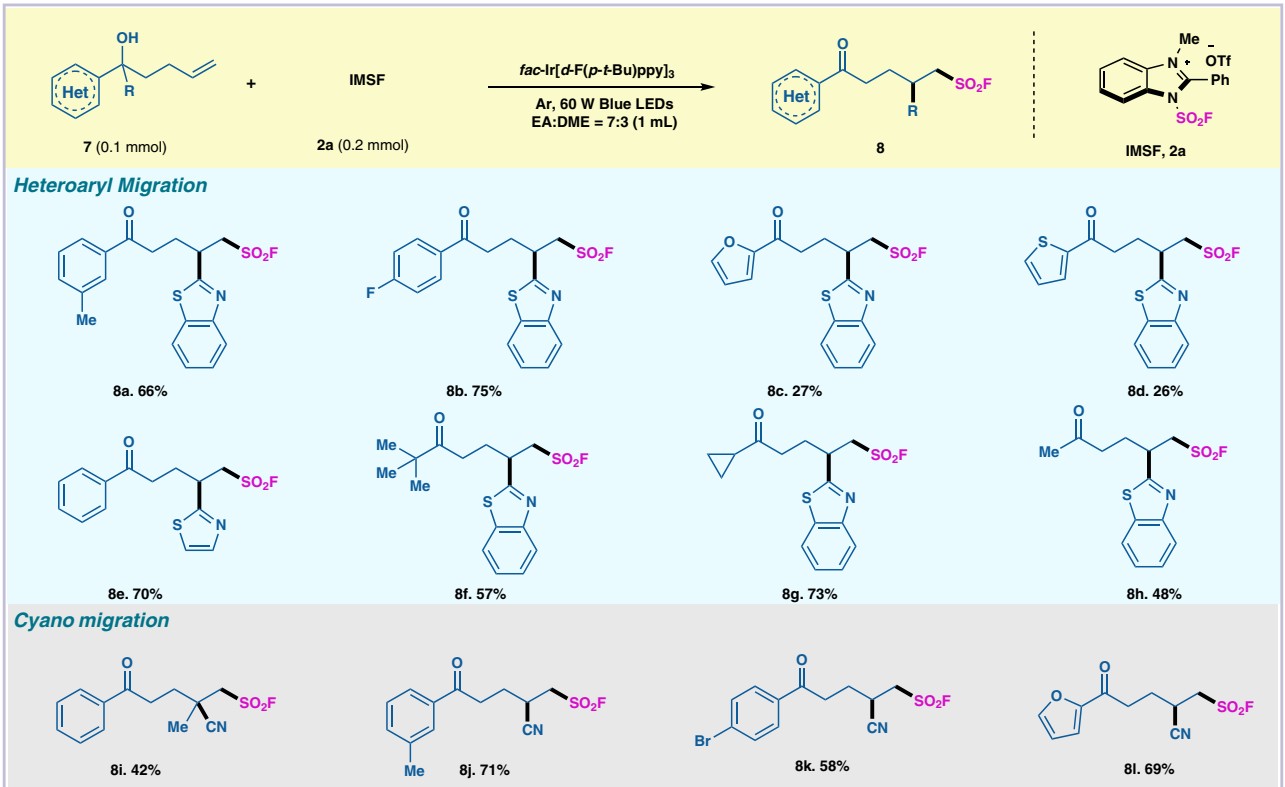

**Fig. 3 Substrate scope of the radical hydrofluorosulfonylation.** All reactions were carried out with olefins **5** (0.10 mmol), IMSF salt **2** (0.20 mmol, 2.0 equiv), *fac*-Ir[*d*-F-(*p*-*t*-Bu)ppy]₃ (2 mol%) and cyclohexa-1,4-diene (1.5 equiv) in 2-methyltetrahydrofuran:Acetone = 9:1 (1 mL) under Ar and 60 W blue LEDs. [a] IMSF reagent **2a** was used. [b] IMSF reagent **2b** was used.

**Fig. 4 Substrate scope of radical migration fluorosulfonylation.** All reactions were carried out with alkenes **7** (0.10 mmol), IMSF salt **2a** (0.20 mmol, 2.0 equiv.), *fac*-Ir[*d*-F(*p*-*t*-Bu)ppy]₃ (2 mol%) in DME:EA = 4:1 (1.0 mL) under Ar and 60 W blue LEDs about 12 h.

**Fig. 5 Synthetic applications and mechanistic studies. a** Cycloaddition of alkenylsulfonyl fluoride. **b, c** SuFEx click reaction of alkenylsulfonyl fluoride **3 m**. **d** Radical inhibition experiment. **e** Free radical clock experiment. **f** Control experiment for the production of Z-alkenylsulfonyl fluoride.

## Methods

**General procedure for the synthesis of sulfonyl fluoride imidazolium salt reagent 2.** Sodium hydride (60% dispersion in mineral oil.) (36 mmol, 1.2 equiv.) was added to corresponding imidazole (30 mmol, 1 equiv.) in dry DMF (100 mL). The mixture was stirred for 1 h; A balloon volume of sulfuryl fluoride gas was then added to the reaction system. The reaction progress was monitored by TLC. After the reaction was completed, the solvent was evaporated in vacuo. Then, the reaction crude was quenched with water and extracted with ethyl acetate (60 mL × 3). The organic layer was dried over $Na_2SO_4$, and evaporated in vacuo. The product was purified by flash column chromatography on silica gel with n-pentane/ethyl acetate as the eluent to give the corresponding 2- aryl-1H-benzo[d]imidazole-1-sulfonyl fluoride. Then a solution of the corresponding 2- aryl-1H-benzo[d]imidazole-1-sulfonyl fluoride in DCM (50 mL) was added dropwise MeOTf (45 mmol) at 0 °C. Then, the mixture was stirred at room temperature for 12 h, while monitoring by TLC. After that time, the mixture was concentrated under rotary evaporation to give a white solid (or a viscous liquid) crude product, to which tert-Butyl methyl ether (30 mL) was added. With vigorous stirring, a solid precipitate was formed. The precipitate was washed with tert-Butyl methyl ether (30 mL × 3) and dried in vacuo to yield the title compound (**2a-2e**) as a white solid.

**General procedure for the synthesis of product 3.** Condition A: Under argon, to a solution of 4CzIPN (2 mol%), $KH_2PO_4$ (2.5 equiv) and IMSF reagent **2b** (0.2 mmol, 2 equiv.) in dried DME (1 mL) was added corresponding alkenes **1** (0.1 mmol) at room temperature. After that, the tube was exposed to a 60 W blue LEDs about 10 h until the reaction was completed as monitored by TLC analysis. The reaction mixture was evaporated in vacuo. The crude products were directly purified by flash chromatography on silica gel to give the desired product.

**General procedure for the synthesis of product 4.** Condition B: Under argon, to a solution of PC 1 (2 mol%), $KH_2PO_4$ (2.5 equiv) and IMSF reagent **2b** (0.2 mmol, 2 equiv.) in dried EA:DME = 4:1 (1 mL) was added corresponding alkenes **1** (0.1 mmol) at room temperature. After that, the tube was exposed to a 60 W blue LEDs about 12 h, then 1.5 ml of acetonitrile containing Ir{[dF(CF₃)ppy]₂(dtbbpy)} PF₆ (2 mol%) was injected into the reaction tube about 12 h until the reaction was completed as monitored by TLC analysis. The reaction mixture was evaporated in vacuo.The crude products were directly purified by flash chromatography on silica gel to give the desired product.

**General procedure for the synthesis of product 6.** Condition C: Under argon, to a solution of fac-Ir[d-F-(p-t-Bu)ppy]₃ (2 mol%), 1,4-cyclohexadiene (1.5 equiv.) and IMSF reagent **2** (0.2 mmol, 2 equiv.) in dried 2-methyltetrahydrofuran:Acetone = 9:1 (1 mL) was added corresponding alkenes **5** (0.1 mmol) at room temperature. After that, the tube was exposed to a 60 W blue LEDs about 12 h, then until the reaction was completed as monitored by TLC analysis. The reaction mixture was evaporated in vacuo. The crude products were directly purified by flash chromatography on silica gel to give the desired product.

**General procedure for the synthesis of product 8.** Condition D: Under argon, to a solution of *fac*-Ir[*d*-F-(*p*-*t*-Bu)ppy]₃ (2 mol%), and IMSF reagent **2a** (0.2 mmol, 2 equiv.) in dried EA:DME = 7:3 (1 mL) was added corresponding alkenes **7** (0.1 mmol) at room temperature. After that, the tube was exposed to a 60 W blue LEDs about 12 h, then until the reaction was completed as monitored by TLC analysis. The reaction mixture was evaporated in vacuo. The crude products were directly purified by flash chromatography on silica gel to give the desired product.

## Data availability

The authors declare that the main data supporting the findings of this study, including experimental procedures and compound characterization, are available within the article and its Supplementary Information files, or from the corresponding author upon request. X-ray structural data of compound **2b** are available free of charge from the Cambridge Crystallographic Data Center under the deposition number CCDC 2164689. These data can be obtained free of charge from The Cambridge Crystallographic Data Center via www.ccdc.cam.ac.uk/data_request/cif.

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

## Acknowledgements

This work was supported by the National Natural Science Foundation of China (Nos. 21971107, and 2201101) and China Postdoctoral Science Foundation (2021T140309 and 2021M691511).

## Author contributions

Y.W. and W.Z. designed and guided this project. H.L. is responsible for the plan and implementation of the experimental work. X.L., Z.Z., M.H., J. L. and X.W. analyzed the data. Y.W. and W.Z. co-wrote the manuscript. S.N. and Y.P. discussed the results and commented on the manuscript.

## Competing interests

The authors declare no competing interests.
