## [Peer Review File · Nature Communications]

REVIEWERS' COMMENTS

Reviewer #1 (Remarks to the Author):

The introduction of SO₂F group into organic molecules has received increasing attention since Sharpless and coworkers developed the SuFEx chemistry. To date, different methods, including radical approach, have been developed in the field. Comparing with the previous reports, the manuscript by Wang can coworkers developed a new reagent benzimidazolium sulfonate (IMSF) for fluorosulfonylation of alkenes through a photoredox catalyzed radical pathway. A variety of Z/E-fluorosulfonylated alkenes were stereoselectively prepared by fine-tuning the reaction conditions. This reagent can also be used for the hydrofluorosulfonylation of alkenes and migration fluorosulfonylation reaction. To demonstrate the synthetic utility, the modification of the complex molecules and the transformations of the resulting fluorosulfonylated products have also been conducted. However, what the referee concerned is the novelty of this work. The preparation of IMSF and its application in radical fluorosulfonylation of alkenes has been developed in 2021, see patent CN 113248444 A. However, the authors did not cite this patent in the paper. Although the current approach describes a variety of transformations of IMSF, the concept is not novel. The paper does not meet the high criteria of the Nature Communications, a special journal might be suitable for it.

Other comments:

1. Explanations on the stereoselective formation of Z- or E-fluorosulfonylated alkenes should be provided.
2. The authors should acknowledge the contribution of the patent in the introduction.
3. The IMSF derived trifluoromethoxylating reagent has also been reported previously, the authors should cite *Angew. Chem. Int. Ed.* 2021, 60, 10211; cite the radical migration fluoroalkylation: *J. Am. Chem. Soc.* 2017, 139, 1388–1391; cite other fluorosulfonyl radical reactions: *Angew. Chem. Int. Ed.* 2021, 60, 27271-27276; *Org. Lett.* DOI: <https://doi.org/10.1021/acs.orglett.2c01336>

Reviewer #2 (Remarks to the Author):

The authors (Wang et al.) have provided a well-written manuscript introducing an air-stable crystalline imidazolium sulfonate cationic salt reagent IMSF. This bench-stable redox-active reagent offers a protocol for the radical fluorosulfonylation of unsaturated hydrocarbons with

good yield and good stereoselectivity. Importantly, showcase its ability to react with various alkenes to produce alkenyl sulfonyl fluoride, alkylsulfonyl fluoride, and migratory fluorosulfonylating product. Additionally, they used this chemistry to achieve late-stage diversifications of Estrone and Vitamin E. That is valuable chemistry. Given the current interest in using SuFEx click chemistry as a connection hub or a pharmacophore warhead, I am sure this manuscript will receive attention in chemical biology and medicinal chemistry. I am therefore supportive of publishing this paper in Nature communication after some revisions.

1, The title of the manuscript is a bit confusing. I agree that this is an applicable synthetic methodology to generate the alkenyl sulfonyl fluoride from some olefins. However, I am not convinced that this methodology will be so general toward the substrates with N-, O-, nucleophiles since the reagent is a strong electrophile.

2, I am not convinced that this methodology could be scalable, as the authors mentioned in the abstract. They used "magic methyl" to generate their reagent while commenting that "the application of N₃SO₂F and ClSO₂F were limited by tedious and hazardous"? The toxicity of MeOTf will be only more significant compared with ClSO₂F and N₃SO₂F. Moreover, removing the excess reagent (2-3eq) and byproduct imidazole will be necessary if one needs to scale up this procedure. It could be tough to do.

Reviewer #3 (Remarks to the Author):

Wang and co-workers describe a new synthesis of sulfonyl fluorides via the formation of F-SO₂ radicals. Sulfonyl fluorides are sort after targets as they are found in wide-ranging applications. Although there are many methods for their synthesis, routes that proceed via the F-SO₂ radical are scarce, due to the difficulty in forming the radical.

The author's solution to this challenge is to use a benzimidazolium salt that can be activated reductively to for the required radical. The salt is an air-stable solid. The reagent can be used in combination with styrenes to form alkenyl sulfonyl fluorides and with simple alkenes to form alkyl sulfonyl fluorides. A decent scope is presented, and the reactions provide the products in moderate to good yields.

This is interesting chemistry, but there are some issues to address.

There are existing methods based on using F-SO₂ radical, which is generated from Cl-SO₂-F. This work is mentioned by the authors, but this needs to be prominently mentioned early in the introduction, not towards the end. This should also be in the abstract. This is how this chemistry is presently achieved; this can not be hidden.

In the synthesis of alkenyl sulfonyl fluorides, the authors split the work into a section described as Z-selective reactions. There are only two examples of selective Z-alkene formation; the remainder of the examples are essentially 1:1. Again, presentation is important, and it is not necessary to mislead with bold claims.

Finally, the claim that sulfonyl fluorides are "...variegated transformations in all facets of chemical science" is simply not true. There is lots of chemistry for which sulfonyl fluorides are not relevant.

This is interesting chemistry. The manuscript needs to be toned down, so that the chemistry is presented clearly, and so that prior work is adequately shown.

Reviewer #1 (Remarks to the Author):

The introduction of SO₂F group into organic molecules has received increasing attention since Sharpless and coworkers developed the SuFEx chemistry. To date, different methods, including radical approach, have been developed in the field. Comparing with the previous reports, the manuscript by Wang and coworkers developed a new reagent benzimidazolium sulfonate (IMSF) for fluorosulfonylation of alkenes through a photoredox catalyzed radical pathway. A variety of Z/E-fluorosulfonylated alkenes were stereoselectively prepared by fine-tuning the reaction conditions. This reagent can also be used for the hydrofluorosulfonylation of alkenes and migration fluorosulfonylation reaction. To demonstrate the synthetic utility, the modification of the complex molecules and the transformations of the resulting fluorosulfonylated products have also been conducted. However, what the referee concerned is the novelty of this work. The preparation of IMSF and its application in radical fluorosulfonylation of alkenes has been developed in 2021, see patent CN 113248444 A. However, the authors did not cite this patent in the paper. Although the current approach describes a variety of transformations of IMSF, the concept is not novel. The paper does not meet the high criteria of the Nature Communications, a special journal might be suitable for it.

Response: We appreciate the referee's comment. However, we believe this work has significant novelty and conceptual difference with the previous reports on radical fluorosulfonylation. The newly developed IMSF reagent is derived from the stable gaseous SO₂F₂, which provides a practical way towards various radical addition reactions compared to the recent advance using ClSO₂F source. Furthermore, this reagent has been patented in 2020 (CN 111187219A), prior to the aforementioned patent of 2021. In order to give fair credit, both patents have been included in the references.

Other comments:

1. Explanations on the stereoselective formation of Z- or E-fluorosulfonylated alkenes should be provided.

Response: Thank you for your suggestion. The control experiments have been performed. The treatment of E-alkenylsulfonyl fluoride 3n with iridium catalyst in the absence of IMSF reagent furnished the Z-alkenylsulfonyl fluoride 4c in 43% yield and recovered 3n in 48% yield (Fig. 5, f). This showed that the generation of Z-alkenylsulfonyl fluoride probably underwent an olefin

isomerization process. The stereoselectivity of Z- or E-fluorosulfonylated alkenes is governed mainly by photocatalysts used.

2.The authors should acknowledge the contribution of the patent in the introduction.

Response: The contribution of the patent was acknowledged in the introduction.

3.The IMSF derived trifluoromethoxylating reagent has also been reported previously, the authors should cite *Angew. Chem. Int. Ed.* 2021, 60, 10211; cite the radical migration fluoroalkylation: *J. Am. Chem. Soc.* 2017, 139, 1388–1391; cite other fluorosulfonyl radical reactions: *Angew. Chem. Int. Ed.* 2021, 60, 27271-27276; *Org. Lett.* DOI: <https://doi.org/10.1021/acs.orglett.2c01336>

Response: Thank you for your comments. The reference has been cited in the revised manuscript.

Reviewer #2 (Remarks to the Author):

Dear Editor,

The authors (Wang et al.) have provided a well-written manuscript introducing an air-stable crystalline imidazolium sulfonate cationic salt reagent IMSF. This bench-stable redox-active reagent offers a protocol for the radical fluorosulfonylation of unsaturated hydrocarbons with good yield and good stereoselectivity. Importantly, showcase its ability to react with various alkenes to produce alkenyl sulfonyl fluoride, alkylsulfonyl fluoride, and migratory fluorosulfonylating product. Additionally, they used this chemistry to achieve late-stage diversifications of Estrone and Vitamin E. That is valuable chemistry. Given the current interest in using SuFEx click chemistry as a connection hub or a pharmacophore warhead, I am sure this manuscript will receive attention in chemical biology and medicinal chemistry. I am therefore supportive of publishing this paper in *Nature communication* after some revisions.

1, The title of the manuscript is a bit confusing. I agree that this is an applicable synthetic methodology to generate the alkenyl sulfonyl fluoride from some olefins. However, I am not convinced that this methodology will be so general toward the substrates with N-, O-, nucleophiles since the reagent is a strong electrophile.

Response: Thank you for your suggestion. The title of the manuscript has been revised to be more appropriate.

2, I am not convinced that this methodology could be scalable, as the authors mentioned in the abstract. They used "magic methyl" to generate their reagent while commenting that "the application of N₃SO₂F and ClSO₂F were limited by tedious and hazardous"? The toxicity of MeOTf will be only more significant compared with ClSO₂F and N₃SO₂F. Moreover, removing the excess reagent (2-3eq) and by product imidazole will be necessary if one needs to scale up this procedure. It could be tough to do.

Response: Thank you for your comments. We have revised the comments of each reagent in the manuscript and we will try to replace MeOTf in the preparation of IMSF reagent.

Reviewer #3 (Remarks to the Author):

Wang and co-workers describe a new synthesis of sulfonyl fluorides via the formation of F-SO₂ radicals. Sulfonyl fluorides are sort after targets as they are found in wide-ranging applications. Although there are many methods for their synthesis, routes that proceed via the F-SO₂ radical are scarce, due to the difficulty in forming the radical.

The author's solution to this challenge is to use a benzimidazolium salt that can be activated reductively to for the required radical. The salt is an air-stable solid. The reagent can be used in combination with styrenes to form alkenyl sulfonyl fluorides and with simple alkenes to form alkyl sulfonyl fluorides. A decent scope is presented, and the reactions provide the products in moderate to good yields.

This is interesting chemistry, but there are some issues to address.

·There are existing methods based on using F-SO₂ radical, which is generated from Cl-SO₂-F. This work is mentioned by the authors, but this needs to be prominently mentioned early in the introduction, not towards the end. This should also be in the abstract. This is how this chemistry is presently achieved; this can not be hidden.

Response: Thank you for your suggestion. We have revised the introduction section and clearly described the related work in the first paragraph of the manuscript.

In the synthesis of alkenyl sulfonyl flourides, the authors split the work into a section described as Z-selective reactions. There are only two examples of selective Z-alkene formation; the remainder of the examples are essentially 1:1. Again, presentation is important, and it is not necessary to mislead with bold claims.

Response: Thank you for your suggestion. We have revised the claims in the main text so as not to mislead the reader.

Finally, the claim that sulfonyl fluorides are "...variegated transformations in all facets of chemical science" is simply not true. There is lots of chemistry for which sulfonyl fluorides are not relevant.

This is interesting chemistry. The manuscript needs to be toned down, so that the chemistry is presented clearly, and so that prior work is adequately shown.

Response: We have revised the abstract as suggested and modified the comments in the manuscript.